# SlimLLM: Accurate Structured Pruning for Large Language Models

**Jialong Guo** [1]  **Xinghao Chen** [1]  **Yehui Tang** [1]  **Yunhe Wang** [1]

## Abstract

Large language models(LLMs) have garnered significant attention and demonstrated impressive capabilities in a wide range of applications. However, due to their enormous computational costs, the deployment and application of LLMs are often severely limited. To address this issue, structured pruning is an effective solution to compress the parameters of LLMs. Determining the importance of each sub-module in LLMs and minimizing performance loss are critical issues that need to be carefully addressed in structured pruning. In this paper, we propose an effective and fast structured pruning method named SlimLLM for large language models. For channel and attention head pruning, we evaluate the importance based on the entire channel or head, rather than merely aggregating the importance of individual elements within a sub-module. This approach enables a more holistic consideration of the interdependence among elements within the sub-module. In addition, we design a simple linear regression strategy for the output matrix to quickly recover performance. We also propose layer-based importance ratio to determine the pruning ratio for each layer. Based on the LLaMA benchmark results, our SlimLLM outperforms other methods and achieves state-of-the-art performance.

## 1. Introduction

Large language models(LLMs) (Achiam et al., 2023; Touvron et al., 2023), currently a popular area of research, have garnered significant attention due to their remarkable achievements in handling complex tasks. These models have not only demonstrated impressive capabilities in natural language processing but have also shown potential in a wide range of applications, from creative writing and language translation to complex reasoning and problem-solving. With the enhancement of model capabilities, there is often a corresponding surge in computational expenses. This often limits the deployment and application of LLMs.

Several techniques have been proposed to address the issue of computational cost in large language models, including model pruning (Ma et al., 2023; Chen et al., 2023), quantization (Frantar et al., 2022; Xiao et al., 2023), and knowledge distillation (Saha et al., 2024). Among these, model pruning stands out as a particularly effective strategy. Model pruning (Anwar et al., 2017) involves the systematic removal of redundant or less important parameters from the model, which not only reduces the model's size but also decreases its computational requirements. Given these benefits, model pruning has emerged as an ideal solution for large language models compression, particularly for deployment in environments with limited resources.

Pruning can be categorized into two main types: unstructured pruning and structured pruning. Unstructured pruning involves removing individual weights from the model based on their magnitude or other criteria, which can lead to a sparse weight matrix. There are several representative works in this field, such as SparseGPT (Frantar & Alistarh, 2023) and Wanda (Sun et al., 2023). These methods are effective in reducing the number of parameters but may not be optimal for hardware acceleration due to the irregular sparsity pattern. Structured pruning, on the other hand, aims to remove channels, attention head, or layers, which results in a more regular and hardware-friendly sparsity pattern. For its better compatibility with existing hardware architectures, structured pruning has attracted much attention.

Considering the enormous computational cost required for training LLMs, the method for LLMs pruning typically employs the strategy of post-training pruning. For instance, LLM-Pruner (Ma et al., 2023) measures the importance of weights based on gradient information and recovers the performance of the pruned model through efficient tuning techniques. Due to the necessity of computing gradients, a significant amount of storage and computational resources is required. To address this issue, LoRAPrune (Zhang et al., 2023) proposes to estimate the importance of weights using LoRA (Hu et al., 2021). LoRAP (Li et al., 2024) designs a gradient-free strategy to prune the unimportant channels,

[1]Huawei Noah's Ark Lab, China. Correspondence to: Xinghao Chen <xinghao.chen@huawei.com>, Yunhe Wang <yunhe.wang@huawei.com>.

*Proceedings of the 42nd International Conference on Machine Learning*, Vancouver, Canada. PMLR 267, 2025. Copyright 2025 by the author(s).

and combines Low-Rank matrix approximation to compress the weights of multi-head self-attention. As LoRAP estimate the importance score of weight by calculating the product of the weight magnitude and the corresponding input activation's L2 norm, it ignore the direction of weight vector when pruning channels. To account for the influence of weight vector direction, we construct the feature space of the output and evaluate the importance of channels within this feature space. For MHA, we employ Pearson similarity to assess the significance of each head, which directly treats the head as a whole for evaluation. Besides, we design a simple yet effective linear regression strategy to recover the performance loss due to pruning. At last, we propose a criterion to determine the pruning ratio for each layer.

Based on the LLaMA series of models, we extensively evaluate the effectiveness of our proposed method. As evidenced by the experimental results obtained from the Commonsense Reasoning datasets, our proposed method achieves 98.7% retention of the original performance on LLaMA-7B when the pruning ratio is set at 20% and outperforms current structured pruning methods.

**Contributions.** In this paper, our contributions can be summarized as follows:

- For the MHA sub-layer, we propose similarity importance for head pruning, which evaluates the entire head importance according to the pearson similarity between original output and output without corresponding head's contribution. Besides, we design a greedy search algorithms to find better combination of heads.

- For the FFN sub-layer, we devise a feature space importance method, which construct the feature space by output activations and assess the importance by considering both direction and magnitude of the channel.

- We find a simple yet effective strategy for performance recover. The method employs linear fitting of the outputs before and after pruning each layer to fine-tune the parameters of the output matrix. Experimental results show that the method can recover the loss caused by pruning at an extremely low cost.

- We design a non-uniform strategy to prune the different layers of LLMs, controlling the pruning ratio of each layer based on the cosine similarity between their input and output.

## 2. Related Work

**Importance score of filters.** The filter is the operational unit for structured pruning. In traditional structured pruning (Liu et al., 2017; Zhuang et al., 2020), scaling factors derived from Batch Normalization or learnable inserted masks

consistently serve as indicators of filter importance. For LLMs pruning, Sheared LLaMA (Xia et al., 2023) adopts a similar approach, compressing the model by learning a set of pruning masks. This method can achieve performance comparable to models of equivalent sizes through sufficient training. To efficiently evaluate the importance of filters, several gradient-based methods have been proposed, such as LLM-pruner and LoRAPrune. LLM-Pruner employs a first-order Taylor expansion to evaluate the importance of weights. LoRAPrune uses the gradient of LoRA parameters to measure the importance of model's parameters. Additionally, LoRAP utilizes (Sun et al., 2023)'s methodology to assess the significance of weights, which achieves gradient-free importance estimation for weights. FLAP (An et al., 2024) proposed a fluctuation-based pruning criterion, taking into account the fluctuations in activation values. Both of these methods calculate the importance score of filters by aggregating the importance scores of weights like summation or L2 Norm. Furthermore, SlimGPT (Ling et al., 2024) incorporates the Optimal Brain Surgeon technique in structured pruning for LLMs, where filters are pruned by minimizing the squared error between the outputs before and after pruning.

**Importance score of layers.** For LLMs pruning, current methods typically employ a uniform ratio for pruning each layer. OWL (Yin et al., 2023) finds that non-uniform layerwise sparsity typically obtain better performance and determine the pruning ratio by Layerwise Outlier Distribution. Meanwhile, several works recognize the notable redundancy across the layers of LLMs, and measure the importance of layers through the cosine similarity between inputs and outputs. Some works (Men et al., 2024; Chen et al., 2024; Muralidharan et al., 2024) apply this strategy, using the cosine similarity to measure layer importance for pruning entire layers of LLMs. SlimGPT highlights that the error introduced during pruning in one layer can accumulate and increase with model depth. To address this, SlimGPT designs an incremental pruning ratio for layers.

**Compression technologies.** Based on the importance score of sub-module, there are different strategies to compression model. Specifically, unstructured pruning removes the least important elements in the matrix, while structured pruning typically removes entire rows (cols) or multiple rows (cols) in the matrix. Moreover, there have been some more coarse-grained pruning strategies recently. Some studies (Men et al., 2024; Kim et al., 2024; Song et al., 2024) compress models by removing entire layers, and others (Chen et al., 2024) replace transformer blocks with a lightweight network. Besides, low-rank approximation (Hsu et al., 2022; Yu & Wu, 2023) serves as an efficient technique for model compression. Low-rank approximation typically transforms matrices in the model into the product of two smaller matrices, thereby reducing the number of parameters. The

method proves to be highly effective when the matrix displays a pronounced low-rank property.

## 3. Preliminary

**Principal component analysis.** Principal component analysis (PCA) is a widely used dimension reduction technique in data analysis. PCA can capture the directions of maximum variance in the original data. Given the data $X \in R^{n \times m}$, where n is the number of data and m represents the dimension of data. PCA requires to calculate the covariance matrix of data, it can be describe as follows:

$$Cov = (X - \mu)^T (X - \mu). \tag{1}$$

where $\mu$ is the mean of data. Then, the covariance matrix can be decomposed into the following form:

$$Cov = Q \Lambda Q^{-1}. \tag{2}$$

where $Q$ is the matrix composed of eigenvectors, $\Lambda$ is the diagonal matrix, with each diagonal element representing an eigenvalue. The eigenvalues represent the magnitude of variance explained by each principal component. To obtain the reduced-dimensional data, the original data is projected onto the selected principal components.

## 4. Methods

We introduce a novel approach to pruning large language models (LLMs). Our method involves compressing model parameters by reducing the number of attention heads and feed-forward network (FFN) channels. Specifically, we develop two distinct criteria for pruning different sub-layers. Then, we employ a linear regression strategy to finetune the weights of output matrix. At the end, we apply our non-uniform pruning ratio for each layer. We show our method in Figure 1.

### 4.1. Similarity Importance for Attention Head Pruning

For multi-head self-attention (MHA), given the input activation $X \in \mathbb{R}^{N \times C}$, where N is the length of sequence and C is the feature dimensions. The multi-head self-attention mechanism can be described as follows:

$$Q_i = XW_Q^i, K_i = XW_K^i, V_i = XW_V^i,$$
$$head_i = Softmax(Q_i K_i^T / \sqrt{d_k}) V_i, \tag{3}$$
$$MHA(X) = \sum_{i=1}^{h} head_i W_O^i.$$

---

**Algorithm 1** Greedy Search for head pruning
 **Input:** pruned attention heads set: $S_p = \{head_i,$ if $head_i$ is pruned.$\}$, unpruned attention heads set: $S_{-p} = \{head_i,$ if $head_i$ is unpruned.$\}$
 **Output:** left attention heads set $S_{left}$
 $S_{left} = S_{-p}$
 $O_{-p} = Output(S_{-p})$
 $O_{all} = Output(S_p + S_{-p})$
 $Sim = Pearson(O_{-p}, O_{all})$
 **for** $h_i \in S_p$ **do**
   **for** $h_j \in S_{-p}$ **do**
     $O = Output(S_{-p} - h_j + h_i)$
     $s = Pearson(O, O_{all})$
     **if** $s > Sim$ **then**
       $Sim = s$
       $S_{left} = S_{-p} - h_j + h_i$
     **end if**
   **end for**
   $S_{-p} = S_{left}$
 **end for**

---

where $W_Q^i, W_K^i, W_V^i$ are query, key and value matrices in $i$-th head, $d_k$ is the dimension of head, $h$ is the number of head, $W_O^i$ is the weight matrix for the final linear projection correspond to $i$-th head. For structured pruning, when the $W_O^i$ is pruned, it effectively means that the $i$-th head's contribution to the final output is eliminated.

As we mentioned above, we consider the influence of $W_O^i$ to evaluate the importance of each head. Given the input of output linear projection $X$, the score of $i$-th head can be described as follows:

$$Score_i = -Pearson(XW_O, XW_O - X_i W_O^i). \tag{4}$$

We use pearson similarity to measure the linear correlation between the original output and the output without contribution of $W_O^i$. Considering the interaction between outputs of each head, we design a greedy search algorithms to maximize similarity. For each pruned head, we replace the unpruned heads in turn and evaluate the pearson similarity. Then we select combinations of heads with maximum similarity as heads left. The algorithm 1 shows detail process of our greedy search for head pruning.

### 4.2. Feature Space Importance for Channel Pruning

When performing channel pruning, it means that we remove a row or column from the weight matrix, which represents a vector. Current methods typically first consider the importance of individual elements in each row (or column), and then measure the importance of individual channels in an aggregated manner. This leads to the information loss of

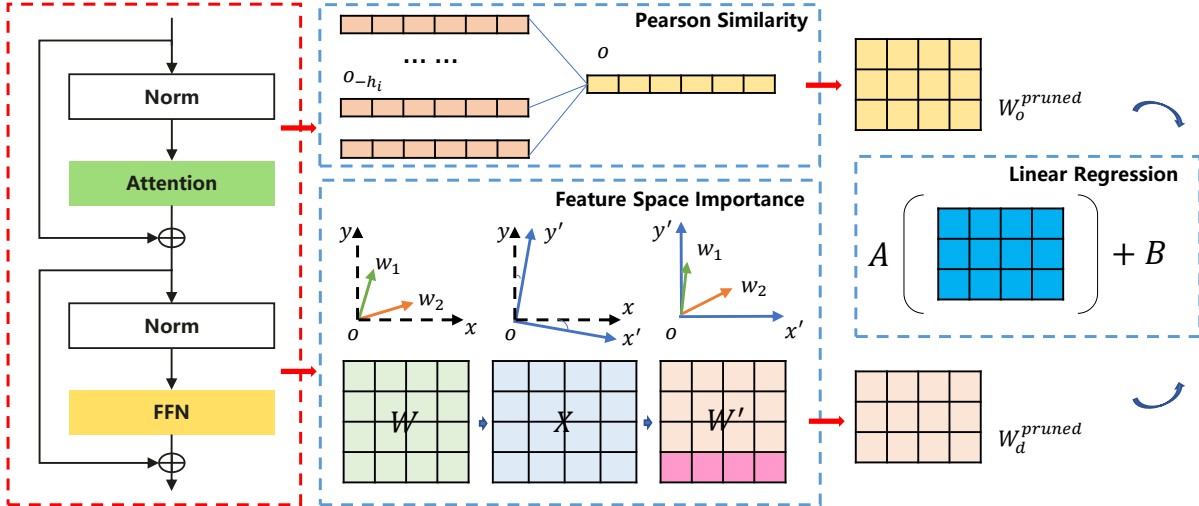

**Figure 1.** The overall framework of our proposed SlimLLM. $o_{-h_i}$ denotes the output excluding the $i$-th head. For the MHA sub-layer, we employ the Pearson similarity between $o_{-h_i}$ and the sum of all heads' output $o$ to evaluate the importance of each head, and prune the head with higher similarity when it is inoperative. For the FFN sub-layer, we map down matrix to the feature space of the output activation, and calculate the channel importance based on the eigenvalues corresponding to the eigenvectors. Finally, we apply linear regression to fine-tune the output matrix of each sub-layer.

vector direction. Different from previous works, we propose a novel method that takes into account both the direction and magnitude of vectors when assessing importance.

Inspired by principal component analysis, we can construct a feature space and evaluate the importance of each direction by calculating the eigenvectors and eigenvalues of the output matrix. Let $Y \in R^{N \times D}$ represents the output of final linear projection, where $N$ is the sequence length and $D$ is the output dimension. We can calculate its eigenvectors and eigenvalues by PCA. Let $Q \in R^{D \times D}$ represents the matrix composed of eigenvectors, $M \in R^D$ represents the vector composed of eigenvalues, and $Q_{:,i}$ represents the eigenvector corresponding to the $i$-th eigenvalue $M_i$. Then, we can map the down matrix $W_{down}$ of FFN to the space composed of eigenvectors using the following formula:

$$W' = W_{down}^T Q, \qquad (5)$$

Combining the eigenvalues corresponding to each eigenvector, the importance score of $i$-th eigenvector can be calculated as follows:

$$C_i = sigmoid(M_i/\bar{M}), \qquad (6)$$

where $\bar{M}$ is the mean of $M$. We use the function of $sigmoid$ to smooth the eigenvalues, allowing channel importance to take into account the influence of more eigenvector directions. Then, the importance score of $j$-th channel based on

feature space can be calculated as follows:

$$I_j^d = ||W'_{j1}C_1, W'_{j2}C_2, ..., W'_{jD}C_D||_2, \qquad (7)$$

where $W'_{ji}$ represents the element in the $j$-th row and $i$-th column of $W'$.

Following (Li et al., 2024), we consider the dependencies between neurons and the influence of input activation. The formulation of $j$-th channel group importance is as follows:

$$I_j = ||X_j||_2 I_j^d + ||X_{L2}W_{gate}^j||_2 + ||X_{L2}W_{up}^j||_2, \qquad (8)$$

where $||X_j||_2$ is the L2 Norm of the input corresponding $j$-th channel. $X_{L2}$ denotes a vector comprising the L2 Norm of the input activations corresponding to $W_{gate}^j$ or $W_{up}^j$. $W_{gate}^j$ and $W_{up}^j$ are the elements in the $j$-th row of gate matrix and up matrix in FFN.

### 4.3. Linear Regression for Performance Recovery

Since we reduce the number of attention heads and channels directly, we adopted a simple and fast linear regression strategy to restore model performance. Let $O$ represents the output of final linear projection of MHA or FFN. The $i$-th dimension of output before and after pruning are respectively expressed as $O_i$ and $O_i^{pruned}$. We apply a simple linear fitting of the output before and after pruning. The

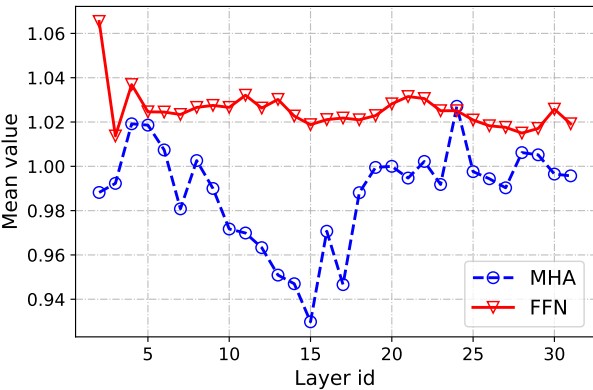

*Figure 2.* Different layers' mean value of the coefficients $A$ in MHA and FFN on LLaMA-7B.

formulation is as follows:

$$O_i = A_i O_i^{pruned} + B_i, \qquad (9)$$

Where $A_i$ and $B_i$ are coefficients of a first-order linear function corresponding $i$-th dimension of output. The coefficients can be calculated using least squares. Additionally, we select the pearson similarity to determine the importance of attention heads, which helps maximize the linear correlation between $O_i$ and $O_i^{pruned}$ and reduces fitting error. To mitigate the computational overhead, we implement our piecewise importance score within the FFN.

After the coefficients of $A$ and $B$ is calculated, the formulation of final linear projection can be described as follows:

$$O = X_{in}(A \cdot W_O)^T + B, \qquad (10)$$

where $X_{in}$ is the input of final linear projection of MHA or FFN.

We list the mean values of $A$ obtained by linear regression for each sub-layer under a pruning ratio of 50% on LLaMA-7B. The results are show in Figure 2. The results reveal that the mean values of $A$ consistently hover around 1.0. This observation suggests that despite a substantial pruning ratio, the output magnitude of the sub-layer remains closely aligned with the original output, thereby preserving the essential characteristics of the model's performance. By making fine adjustments to the output matrix through linear regression, the model can reduce the output error caused by pruning and effectively maintain its performance.

Simultaneously, the values of $B$ also fluctuate around zero.

### 4.4. Pruning Ratio for Layers

Finding an appropriate pruning ratio for each layer is crucial. Maintaining a uniform pruning ratio across all layers often

leads to suboptimal outcomes. Inspired by the work in (Men et al., 2024), which suggests that the impact of a transformer block is determined by its ability to change hidden states, we utilize the cosine similarity between the input and output of a layer to determine the pruning ratio.

Let $X_{i,t}$ represents the $t$-th row of input to layer $i$. The pruning ratio of layer $i$ is calculated by follows:

$$r_i^{layer} = r_0 \cdot softmax(\alpha \cdot E \frac{X_{i,t}X_{i+1,t}^T}{||X_{i,t}||_2 \cdot ||X_{i+1,t}||_2}), \ (11)$$

Both of $r_0$ and $\alpha$ are constant and their value are related to the pruning ratio. E represents the operation of expectation. Additionally, for smaller pruning ratios, a larger $\alpha$ can be used, whereas for larger pruning ratios, $\alpha$ should be reduced.

Based on experimental observations, the cosine similarity between the first layer and the last layer is found to be relatively low. This observation is in accordance with the prevailing notion that the first and last layers tend to be of greater significance. Similar to many current methods (Ma et al., 2023; Kim et al., 2024), we skip the first and last layers during pruning. Moreover, at a pruning ratio of 20%, we adopt the strategy from LLM-pruner and bypass additional layers to achieve higher model performance. In the case of the LLaMA-7B model, it is observed that the cosine similarity exhibits a gradual upward trend as the layer depth increases. Consequently, we employ a stratified pruning strategy, wherein the pruning ratios are systematically lower in the shallower layers and incrementally higher in the deeper layers.

## 5. Experiments

### 5.1. Experimental Settings

**Evaluation and Datasets.** We evaluate the performance of model by performing zero-shot task classification on common sense reasoning datasets, which follow the setting of LLM-pruner (Ma et al., 2023), including BoolQ (Clark et al., 2019), PIQA (Bisk et al., 2020), HellaSwag (Zellers et al., 2019), WinoGrande (Sakaguchi et al., 2021), ARC-easy (Clark et al., 2018), ARC-challenge (Clark et al., 2018) and OpenbookQA (Mihaylov et al., 2018). Meanwhile, a zero-shot perplexity (PPL) evaluation is also conducted on the WikText2 (Merity et al., 2016) and PTB datasets (Marcus et al., 1993). We performed extensive experiments on LLaMA-1 and LLaMA-2 models to rigorously validate the efficacy of our proposed method.

**Implementation Details.** For the calculation of importance score, we randomly selected 32 samples from Bookcorpus, and the sequence length of each samples is 128. When calculating the pruning ratio of layers, we skipped some of

*Table 1.* Zero-shot performance of the compressed LLaMA-7B. The average score is computed on the Commonsense Reasoning datasets. The **"bolded"** represents the best result under the same pruning ratio.

| Ratio | Method | WikiText2 | PTB | BoolQ | PIQA | HellaSwag | WinoGrande | ARC-e | ARC-c | OBQA | Average |
|---|---|---|---|---|---|---|---|---|---|---|---|
| Ratio=0% | Llama-7B | 12.62 | 22.14 | 73.18 | 78.35 | 72.99 | 67.01 | 67.45 | 41.38 | 42.4 | 63.25 |
| | LLM-pruner | 19.09 | 34.21 | 57.06 | 75.68 | 66.8 | 59.83 | 60.94 | 36.52 | 40.0 | 56.69 |
| Ratio=20% | LoRAPrune | 20.67 | 34.12 | 57.98 | 75.11 | 65.81 | 59.90 | 62.14 | 34.59 | 39.98 | 56.50 |
| w/o tune | LoRAP | **15.69** | **25.86** | 71.93 | **76.44** | **69.98** | 65.9 | 60.56 | 38.48 | **40.4** | 60.53 |
| | Ours | 15.95 | 26.09 | **72.72** | 75.95 | 69.82 | **66.06** | **64.48** | **39.33** | 40.2 | **61.22** |
| | LLM-pruner | 17.58 | 30.11 | 64.62 | 77.2 | 68.8 | 63.14 | 64.31 | 36.77 | 39.8 | 59.23 |
| Ratio=20% | LoRAPrune | 16.80 | 28.75 | 65.62 | **79.31** | 70.00 | 62.76 | 65.87 | 37.69 | 39.14 | 60.05 |
| w/ tune | LoRAP | 16.35 | 27.06 | 72.94 | 76.93 | 70.9 | 65.75 | 64.31 | 39.93 | **41.2** | 61.7 |
| | Ours | **15.55** | **26.66** | **74.71** | 76.61 | **71.23** | **66.54** | **66.96** | **40.61** | 40.2 | **62.41** |
| | LLM-pruner | 112.44 | 255.38 | 52.32 | 59.63 | 35.64 | 53.20 | 33.50 | 27.22 | 33.40 | 42.13 |
| Ratio=50% | LoRAPrune | 121.96 | 260.14 | 51.78 | 56.90 | 36.76 | 53.80 | 33.82 | 26.93 | 33.10 | 41.87 |
| w/o tune | LoRAP | 56.96 | 87.71 | 57.8 | 63.82 | 46.96 | 57.3 | 40.36 | 27.73 | 36.80 | 47.25 |
| | Ours | **37.89** | **67.68** | **63.33** | **65.40** | **49.94** | **58.80** | **45.83** | **30.38** | **37.00** | **50.10** |
| | LLM-pruner | 38.12 | 66.35 | 60.28 | 69.31 | 47.06 | 53.43 | 45.96 | 29.18 | 35.60 | 48.69 |
| Ratio=50% | LoRAPrune | 30.12 | 50.30 | 61.88 | **71.53** | 47.86 | 55.01 | 45.13 | 31.62 | 34.98 | 49.71 |
| w/ tune | LoRAP | 30.90 | 48.84 | **63.00** | 69.64 | 54.42 | 58.41 | 51.94 | 32.00 | 35.80 | 52.17 |
| | Ours | **26.71** | **42.19** | 62.78 | 68.99 | **54.73** | **61.01** | **54.55** | **33.28** | 36.80 | **53.16** |

the most important layers. For example, the pruning ratio of first and last layer is set to zero when pruning model. when finetuning, we use a single GPU with 2 epochs on cleaned version of Alpaca (Taori et al., 2023), retaining the same settings as LLM-pruner. We finetune the pruned model with LoRA. The learning rate is set to 1e-4, and the batch-size is 64. For the pruning ratio assigned to each layer, when the pruning ratio is set at 20%, the parameter $\alpha$ in Equation 11 is configured to be 10. When the pruning ratio is increased to 50%, we correspondingly decrease the parameter value, setting it to 7.

**Baselines.** We select the following high-performance structured pruning methods in recent years as benchmarks:

- LLM-pruner (Ma et al., 2023), as the first structured pruning method applied to LLMs, has been widely used as a benchmark in numerous studies. This work groups the structures based on the dependencies between weights and prunes the least important groups using Taylor expansion.

- LoRAPrune (Zhang et al., 2023) is the method which reduces the gradient computation and memory consumption of LLM-pruner by employing the weights of LoRA.

- LoRAP (Li et al., 2024) employs distinct compression strategies for MHA and FFN components within the Transformer architecture. Specifically, it utilizes low-

rank approximation to compress the weight matrices of MHA, while adopting a group importance strategy for FFN. Based on this strategy, the method achieves state-of-the-art performance.

### 5.2. Zero-shot Performance.

Table 1 shows the zero-shot performance of the pruned model. On the LLaMA-7B model, employing a 20% pruning ratio without post-training results in an average score drop of 2.03%. However, through efficient post-training, the accuracy of the pruned model can be enhanced by 1.19%. Compared with other methods such as LoRAP, the score of our method is improved by 0.7% with 20% pruning ratio. When compared with PPL, LoRAP demonstrates a slight advantage in performance on the WikiText2 dataset without finetuning. Conversely, it shows a minor deficiency in performance on PTB dataset. In the context of finetuning, our approach attains optimal results on both PPL metrics. When the pruning ratio reaches 50%, our method still manages to retain more performance comparing to other methods. Specifically, in the absence of finetuning, our method attains an average accuracy on the Commonsense Reasoning datasets that surpasses that of the LoRAP by 2.85%. In the finetuning scenario, our method achieves a performance improvement of 1% over other approaches. Specifically, our approach demonstrates superior performance across the majority of Commonsense Reasoning datasets.

Table 2 presents our evaluation results on the LLaMA2-7B

*Table 2.* Zero-shot performance of the compressed LLaMA2-7B. The average score is computed on the Commonsense Reasoning datasets. The **"bolded"** represents the best result under the same pruning ratio.

| Ratio | Method | WikiText2 | PTB | BoolQ | PIQA | HellaSwag | WinoGrande | ARC-e | ARC-c | OBQA | Average |
|---|---|---|---|---|---|---|---|---|---|---|---|
| Ratio=0% | Llama2-7B | 12.18 | 47.25 | 71.04 | 78.40 | 72.96 | 67.17 | 69.32 | 40.53 | 40.80 | 62.89 |
| Ratio=20% | LoRAP | **15.02** | 58.44 | 69.24 | **76.39** | **69.15** | **65.11** | 61.99 | 35.58 | 38.60 | 59.44 |
| w/o tune | Ours | 15.70 | **56.33** | **69.79** | 76.28 | 68.88 | 63.54 | **65.74** | **39.08** | **39.80** | **60.44** |
| Ratio=20% | LoRAP | **14.67** | 57.52 | 70.89 | **78.13** | 69.93 | **65.67** | 65.99 | 38.48 | **39.60** | 61.24 |
| w/ tune | Ours | 15.28 | **55.46** | **72.29** | 78.02 | **70.95** | 64.88 | **67.17** | **38.99** | **39.60** | **61.70** |
| Ratio=50% | LoRAP | 60.89 | 282.22 | 61.86 | 62.23 | 43.98 | **55.41** | 38.51 | 27.65 | 33.00 | 46.09 |
| w/o tune | Ours | **38.64** | **141.06** | **62.69** | **64.74** | **45.91** | 53.28 | **39.73** | **29.01** | **33.2** | **46.93** |
| Ratio=50% | LoRAP | **26.26** | 101.22 | 63.27 | **70.78** | **55.14** | **57.85** | 52.15 | 30.97 | 36.00 | **52.31** |
| w/ tune | Ours | 27.29 | **88.28** | **64.19** | 69.04 | 53.60 | 55.33 | **52.53** | **32.08** | **37.40** | 52.02 |

*Table 3.* Inference latency of compressed model.

| Ratio | #Params | Prefill(s) | decoding(s) |
|---|---|---|---|
| 0% | 6.7B | 0.3008 | 13.12 |
| 20% | 5.4B | 0.1429 | 11.48 |
| 50% | 3.4B | 0.1034 | 9.38 |

*Table 4.* Ablation Studies for the impact of our strategies at 50% pruning ratio.

| Method | WikiText2 | PTB | Avg. |
|---|---|---|---|
| SlimLLM | 37.89 | 67.68 | 50.10 |
| w/o Greedy search | 40.89 | 82.60 | 48.35 |
| w/o feature space importance | 38.26 | 71.21 | 49.74 |
| w/o non-uniform pruning ratio | 66.37 | 123.52 | 42.16 |

model. We compare our method with LoRAP. As can be seen from the results, our method performs better on the majority of datasets. At a pruning ratio of 20%, our method yields a slightly higher PPL on the WikiText2 dataset compared to LoRAP, while achieving better results on the PTB dataset and a greater average score on Commonsense Reasoning datasets. At a 50% pruning ratio, without fine-tuning, our method attains an average score of 46.93%, which is approximately 0.8% higher than LoRAP. After finetuning, LoRAP's average results are slightly higher, which is primarily due to the inability to align the finetuning strategies.

### 5.3. Latency of compressed model

Table 3 illustrates the model size and inference latency of the pruned LLaMA-7B. The prefill latency was evaluated using an input sequence length of 256 and a single-token generation task. To test decoding latency, we set the batch size to 8 and measured the latency for generating 256 tokens. For each specified pruning ratio, the experiment was repeated 20 times to ensure statistical robustness, and the average latency was computed as the definitive outcome. All latency measurements were conducted on a single NVIDIA V100 GPU. As detailed in Table 3, at a pruning ratio of 50%, the latency for generating a single token, given a consistent input sequence length of 256, decreased from 0.3008 seconds to 0.1034 seconds. Simultaneously, the decoding latency is reduced by 28.5% compared to the unpruned model when the pruning ratio is set at 50%.

### 5.4. Ablation Study

To verify the effectiveness of our proposed method, we conducted experiments on the LLaMA-7B model to validate each of the strategies we introduced. Unless otherwise specified, all experiments were conducted without fine-tuning at a pruning ratio of 50%.

**Greedy search for head pruning.** From Table 4, it can be observed that without greedy search for head pruning, the model's performance metrics generally decline across all evaluation criteria. Among these results, the average score of Commonsense Reasoning datasets dropped by nearly 2%, and the PPL metric of WikiText2 and PTB also deteriorated significantly. This indicates that there is interdependence among the outputs of different heads, and the linear correlation of outputs before and after pruning can effectively assess the importance of head combinations. Building on the basis of individual head importance, greedy search can rapidly and effectively identify superior head combinations, thereby enabling the pruned model to retain more of its original performance.

It is worth noting that, for FFN sub-layer, the outputs of different channels also exhibit direct interdependencies. However, considering that the number of channels in the intermediate layer of FFN is considerably larger than the number of attention heads, and given the associated computational overhead, we refrained from employing the greedy search strategy in the channel pruning.

*Table 5.* Ablation Studies for the value of $\alpha$.

| Value | WikiText2 | PTB | Avg. |
|---|---|---|---|
| $\alpha = 1$ | 54.17 | 96.57 | 44.42 |
| $\alpha = 4$ | 43.36 | 83.91 | 44.95 |
| $\alpha = 7$ | 37.89 | 67.68 | 50.10 |
| $\alpha = 10$ | 44.82 | 75.80 | 46.86 |

*Table 6.* Ablation Studies for the impact of our Linear Regression for output matrix.

| Ratio | Method | WikiText2 | PTB | Avg. |
|---|---|---|---|---|
| Ratio=20% | w/ LR | 15.95 | 26.09 | 61.22 |
| w/o tune | w/o LR | 16.78 | 26.98 | 60.89 |
| Ratio=50% | w/ LR | 37.89 | 67.68 | 50.10 |
| w/o tune | w/o LR | 48.66 | 81.64 | 46.66 |

**The effective of feature space importance.** Given that the output values are linear combinations of the corresponding weight vectors of each channel, we assess the importance of each channel based on the feature distribution of the output values. The results demonstrate that feature space importance enhances the selection of pruning channels for the FFN by integrating the significance of individual feature directions. Benefiting from the incorporation of feature space importance, the pruned model achieves an average score improvement of nearly 0.4%, while also realizing notable enhancements in PPL results. The method primarily enhances the original approach based on the magnitude of weight vectors by incorporating the consideration of directional importance. When the magnitude of the weight vectors is large, this strategy can combine vector direction to further optimize the selection of pruning channels.

**The pruning ratio for different layers.** As indicated in Table 4, the pruning ratio assigned to each individual layer exerts the most substantial influence on the model's overall performance. By incorporating layer importance into the pruning ratio allocation, the proposed strategy results in a pruned model with an average performance improvement of approximately 8 percentage points over uniform pruning, significantly alleviating the accuracy drop induced by model compression. Moreover, under the non-uniform pruning strategy, the pruned model's PPL of WikiText2 and PTB is nearly halved. In Equation 11, we introduce the parameter $\alpha$ to control the fluctuation range of the pruning ratios across layers. The Table 5 presents the impact of different values on model performance. As the results indicate, a pruning ratio of 50% combined with $\alpha = 7$ leads to improved model performance after pruning.

**Linear regression for output matrix.** Linear regression is the strategy we employ to rapidly restore the accuracy of pruned models. We perform individual linear fitting for each output dimension of the final output matrices of MHA and FFN within the Transformer blocks, aiming to make the model's output as close as possible to the output before pruning. As shown in Table 6, when the pruning ratio is 20%, the average score of the model without the linear regression strategy is 60.89, which is approximately 0.3% lower than that of the method using this strategy. When the pruning ratio reaches 50%, the linear regression strategy becomes even more effective in enhancing model performance, resulting in an approximate 3.4% improvement compared to

models not using this strategy. The changes in PPL exhibit a similar pattern, with the recovery becoming increasingly significant as the pruning ratio is elevated. For example, at a pruning ratio of 20%, the PPL on the WikiText2 dataset is decreased by 0.83 via the linear regression strategy. When the pruning ratio is increased to 50%, the reduction in PPL reaches 10.77. This indicates that our linear regression strategy is effective in restoring performance under high pruning ratios. Moreover, since we only utilize a small calibration set for the least squares linear regression, the computational cost of this method is extremely low. Since it is only applicable when the output dimensions are identical, we currently employ the linear regression strategy solely on the output matrices of MHA and FFN. Further exploration is needed for rapid fitting of other parameter matrices.

## 6. Conclusion

In this paper, we present SlimLLM, a novel and efficient pruning technique specifically designed for large-scale models. For both the Multi-Head Attention (MHA) and Feed-Forward Network (FFN) sub-layers, we adopt a holistic importance strategy that goes beyond mere aggregation of individual weight importance. Specifically, for head pruning, we directly assess the importance of each head by establishing the linear relationship between the output of a single head and the combined output of all heads. Furthermore, acknowledging the interdependencies among head outputs, we introduce a greedy search algorithm to explore more optimal head pruning combinations. For channel pruning, we integrate both the magnitude and direction of each channel's corresponding weight vector and utilize Principal Component Analysis (PCA) to determine the importance of each feature direction in the output matrix, thereby deriving a feature space importance that is based on both magnitude and direction. In addition, we devise a simple linear regression strategy for the output matrix to rapidly and effectively restore model performance. Considering that different layers of the model have varying degrees of parameter redundancy, we utilize the cosine similarity between the input and output of each layer to determine the pruning ratio for that layer. This approach further enhances the pruned model's performance. Results show that SlimLLM achieves excellent performance across different models and pruning ratios.

## Impact Statement

This paper presents work whose goal is to advance the field of Machine Learning. There are many potential societal consequences of our work, none which we feel must be specifically highlighted here.

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

## A. Quantity of calibration data

Table 7 illustrates the influence of calibration set size on model accuracy. As shown in the results, perplexity tends to decrease with an increasing calibration set size. Notably, when the size is below 16, the average score on Commonsense Reasoning datasets shows considerable variation. The optimal performance is observed when the calibration set size is set to 32. Accordingly, we adopt 32 as the default calibration set size in our pruning framework.

*Table 7.* Ablation Studies for the quantity of calibration data. **Avg.** presents the average score on Commonsense Reasoning datasets.

| Number | 1 | 4 | 8 | 16 | 32 | 64 |
|---|---|---|---|---|---|---|
| WikiText2↓ | 58.8 | 45.89 | 45.98 | 44.3 | 37.89 | 39.56 |
| PTB↓ | 129.95 | 99.64 | 99.25 | 80.06 | 67.68 | 72.05 |
| Avg.↑ | 46.50 | 49.14 | 47.89 | 45.06 | 50.10 | 49.31 |

## B. Experiments on other models.

In addition, we present pruning results on the Vicuna-7B and LLaMA-13B models. The corresponding experimental results are shown in Table 8 and Table 9, respectively. The results demonstrate that our method achieves competitive performance across various pruning ratios. Moreover, larger models tend to suffer less performance degradation under the same pruning ratio. Notably, compared with LoRAP, their fine-tuned models exhibit better performance in some cases under a 50% pruning ratio. We consider this is due to the fact that LoRAP reconstructs the decomposed matrices in MHA during training, thereby enabling the utilization of more parameters in the optimization process.

*Table 8.* Zero-shot performance of the compressed Vicuna-7B.

| Method | Ratio | WikiText2 | PTB | BoolQ | PIQA | HellaSwag | WinoGrande | ARC-e | ARC-c | OBQA | Average |
|---|---|---|---|---|---|---|---|---|---|---|---|
| Vicuna-7B | Ratio=0% | 16.23 | 58.12 | 75.69 | 77.91 | 71.04 | 67.80 | 68.98 | 40.70 | 42.20 | 63.47 |
| w/o tune | Ratio=20% | 20.24 | 68.75 | 74.92 | 76.12 | 67.98 | 65.82 | 67.09 | 39.33 | 42.60 | 61.98 |
| | Ratio=50% | 43.96 | 131.49 | 61.31 | 67.25 | 48.91 | 56.35 | 48.48 | 31.06 | 36.00 | 49.91 |
| w/ tune | Ratio=20% | 17.15 | 59.03 | 76.15 | 76.39 | 69.32 | 64.72 | 68.56 | 39.25 | 41.80 | 62.31 |
| | Ratio=50% | 29.98 | 87.93 | 62.60 | 69.53 | 53.66 | 57.77 | 55.30 | 31.91 | 37.40 | 52.60 |

*Table 9.* Zero-shot performance of the compressed LLaMA-13B.

| Method | Ratio | WikiText2 | BoolQ | PIQA | HellaSwag | WinoGrande | ARC-e | ARC-c | OBQA | Average |
|---|---|---|---|---|---|---|---|---|---|---|
| LLaMA-13B | Ratio=0% | 11.58 | 68.47 | 78.89 | 76.24 | 70.09 | 74.58 | 44.54 | 42.00 | 64.97 |
| w/o tune | Ratio=20% | 13.35 | 74.13 | 77.53 | 74.73 | 69.30 | 70.45 | 42.32 | 41.00 | 64.21 |
| | Ratio=50% | 25.64 | 62.31 | 72.20 | 60.84 | 60.62 | 55.60 | 33.87 | 37.80 | 54.75 |

