# OpenReview forum: "SlimLLM: Accurate Structured Pruning for Large Language Models"
_ICML.cc/2025/Conference — ICML 2025 poster_

### Official Review · Reviewer_JstE · 2025-03-08

**Overall Recommendation:** 3

**Summary:**

This work involves compressing LLMs by width pruning of sublayers in transformer blocks, where MHA and FFN are treated differently. For MHA, head pruning is performed based on the similarity between outputs with and without specific heads. For FFN, channel pruning is carried out using a Wanda-based metric from LoRAP, which additionally considers feature direction along with feature magnitude. Performance recovery is achieved through a linear regression method, and a non-uniform pruning ratio is applied across different blocks. The experiments are exclusively conducted on the LLaMA family.

**Claims And Evidence:**

This work is largely supported by empirical evidence, and the approach to pruning MHA and FFN differently seems interesting. Although additional hyperparameters are introduced, the type of performance recovery employed is novel to me.

However, some aspects of FFN pruning remain unclear, and heuristics are used to determine non-uniform pruning ratios. The paper introduces one method for MHA pruning (A), another for FFN pruning (B), and a third for performance recovery (C). While (A) and (C) appear promising and interesting, the rationale and effectiveness of (A), (B), and (C) independently are not well justified. Additionally, several necessary ablation studies are missing.

**Essential References Not Discussed:**

* The discussion lacks mention of certain pruning studies such as width pruning techniques like FLAP and Minitron, and depth pruning methods including Shortened LLaMA, SLEB, and Minitron. It seems essential to compare this work with FLAP, given its significant benefit of eliminating the need for retraining, and because analyzing the differences between original and pruned features is relevant.
  - [FLAP] https://arxiv.org/abs/2312.11983
  - [Minitron] https://arxiv.org/abs/2408.11796
  - [Shortened LLaMA] https://arxiv.org/abs/2402.02834
  - [SLEB] https://arxiv.org/abs/2402.09025

**Experimental Designs Or Analyses:**

* Measuring the impact of per-head removal as shown in Equation (4) is intriguing, yet its effectiveness is still in question. Regarding MHA pruning, is it more effective than gradient-based importance measures like those used in LLM-Pruner, AWSVD in LoRAP, or the fluctuation metric in FLAP, when all other conditions are identical? Although Tables 1 and 2 offer comparisons with several methods, the inclusion of other factors, such as FFN pruning and non-uniform ratios, makes it challenging to clearly evaluate the effectiveness of the proposed metric for MHA.
  - [FLAP] https://arxiv.org/abs/2312.11983

* Furthermore, the computations in Algorithm 1 appear to be intensive due to its greedy search nature. How long do these computations typically take, and is it possible to provide comments on this aspect?

* I think the pruning importance metric for the FFN is quite similar to LoRAP, except it employs a sigmoid function for the down projection. Why is the down projection designed in this particular way while the gate and up projection do not follow the same design? Is it impossible to apply Eqn (7) to the gate and up projection matrices, or is there another reason?

* Linear regression for performance recovery seems effective; however, are A and B in Eqn (10) additional, newly introduced parameters? It appears that A and B are necessary for the inference process. If this is the case, it may not integrate well with other inference engines, such as vLLM and TensorRT-LLM, because it requires modifications to the forward code and involves saving additional parameters. This could limit the practicality of the work.

* How did the authors set r_0​ and alpha in Eqn (11)? The statement, 'Consequently, we employ a stratified pruning strategy, wherein the pruning ratios are systematically lower in the shallower layers and incrementally higher in the deeper layers,' (line 241 of page 5) seems entirely heuristic and lacks a systematic foundation. Furthermore, the generalizability to other LLMs beyond the LLaMA family is not guaranteed.

* For the results for Llama-2 in Table 2, I am not convinced that SlimLLM outperforms LoRAP. The numerical results, such as WikiText2 PPL and downstream accuracy, appear very similar, and in some cases, LoRAP seems to perform better.

* The number and type of calibration samples have not been investigated.

* The experimental validation in this study seems restricted since only models from the LLaMA family are used. To better ascertain the widespread applicability and superiority of this method, I suggest conducting experimental comparisons with additional models such as OPT, Qwen, Phi, and MoE-based architectures.

* The tests were limited to models with only 7B parameters. Including models with at least 13B parameters, as done in previous studies, is essential to provide a comprehensive assessment.

* The methodology used to calculate latency gains in Table 3 needs further elaboration, especially since width pruning often faces challenges in achieving actual speedups in setups like ZipLM and Shortened LLaMA. Could you clarify which framework was used (e.g., Vanilla HuggingFace, TensorRT-LLM, vLLM)? It's also important to know if the method can accelerate both the prefill and decoding stages, as the current results seem to focus solely on prefill with single-token generation. Is a batch size of 1 appropriate?

**Methods And Evaluation Criteria:**

The proposed framework aligns well with previous studies on LLM pruning. Given the emphasis on LLM compression in this work, the use of evaluation metrics seems adequate.

**Other Comments Or Suggestions:**

* I am uncertain about the relevance of the quantization and low-rank approximation methods discussed in the 'Compression technologies' part of Sect 2. Related Work to this study, as the authors do not test the compatibility of these methods with the proposed pruning technique. Additionally, there appears to be no citation for the studies mentioned in this subsection.
* Figure 3, which illustrates the impact of each layer on LLMs, presents a concept that is already familiar in this field, as evidenced by similar graphs in ShortGPT, SLEB, and Shortened LLaMA. This raises questions about the value this graph adds to the paper. Furthermore, relevant citations are missing in line 274 on page 5, where the paper states, 'Similar to many current methods, we...'.

**Other Strengths And Weaknesses:**

N/A

**Questions For Authors:**

Please refer to the sections on <Claims And Evidence> and <Experimental Designs Or Analyses>.

I like several aspects of this work and appreciate the authors’ efforts in compressing LLMs. However, several justifications and ablation studies appear to be missing, and the experimental validation seems weak compared to relevant papers.

**Relation To Broader Scientific Literature:**

Reducing computational requirements for LLMs is widely discussed, and this work tackles it in a clear and effective manner.

**Theoretical Claims:**

While the paper does not make theoretical claims, it does provide clear descriptions of the equations used in the method.

---

> ### Author Rebuttal · Authors · 2025-04-01
>
> Q1: Proposed metric for MHA
>
> A1: We design similarity-based approach to enhance linear fitting by maximizing output similarity. To verify the effectiveness of our similarity-based approach, we employed the fluctuation metric from FLAP to evaluate the importance of heads, while aligning other configurations. From the results, It can be observed that our similarity-based strategy achieves a lower PPL.
> |    method              | ratio | wikitext2$\downarrow$ |  PTB$\downarrow$  |   Avg.$\uparrow$  |
> |------------------------|-------|-----------|-------|------------------|
> | fluctuation metric     | 50%   | 46.34     | 87.25 |  44.44           |
> | similarity-basd metric | 50%   | 37.89     | 67.68 |  50.10           |
>
> Q2: Computations in Algorithm 1
>
> A2: The computational cost of greedy search is acceptable. The time spent on greedy search is approximately 2 seconds for each layer on Llama-7B.
>
> Q3: Importance metric for the FFN
>
> A3: Yes, there is another reason. When pruning the input channels of the down projection, each channel's input activation only affects the magnitude of its output vector, while the direction is determined by the weights. For up and gate projection, their outputs correspond to the input activations of the down projection, and thus, we have not considered their impact on the direction of the final output matrix.
>
> Q4: Linear regression
>
> A4:  A and B can be integrated into the output projection. Here, the weight matrix of the output projection is $A \cdot W_{o}$, and the bias term is $B + B_{o}$.
>
> Q5: r_0​ and alpha
>
> A5: Under the pruning ratio $r$ r_0 is calculated according to the following formula:
> $ r_{0} = r * $ pruned_layers / total_layers
>
> For alpha, we provide results for various values. Other models can use these results for adjustment.
> |    alpha    | ratio | wikitext2 |  PTB  |   Avg.$\uparrow$  |
> |-------------|-------|-----------|-------|------------------|
> |     1       |  50%  |    54.17  | 96.57 | 44.42            |
> |     4       |  50%  |    43.36  | 83.91 | 44.95            |
> |     7       |  50%  |    37.89  | 67.68 | 50.10            |
> |     10      |  50%  |    44.82  | 75.80 | 46.86            |
>
> Q6: Performance
>
> A6: The experimental results show that the advantages of our model generally decrease after fine-tuning. This is mainly because the fine-tuning strategies of the two methods cannot be fully aligned. During compression, LORAP uses low-rank decomposition for MHA's linear layers, while LORA fine-tuning does not. LORAP reconstructs the decomposed matrices into a single matrix to keep MHA's parameter count unchanged. This increases the parameter count during LORA training.
>
> Q7: Calibration samples
>
> A7: We have added experiments to investigate the impact of the calibration set size on model performance.
> |    alpha    | ratio | wikitext2 |  PTB  |   Avg.$\uparrow$  |
> |-------------|-------|-----------|-------|------------------|
> |     1       |  50%  |    58.8   | 129.95| 46.50            |
> |     4       |  50%  |    45.89  | 99.64 | 49.14            |
> |     8       |  50%  |    45.98  | 99.25 | 47.89            |
> |     16      |  50%  |    44.3   | 80.06 | 45.06            |
> |     32      |  50%  |    37.89  | 67.68 | 50.10            |
> |     64      |  50%  |    39.56  | 72.05 | 49.31            |
>
>
> Q8: Experiments
>
> A8: To demonstrate the generalizability of our strategy, we have included experimental results on the Vicuna-7B.
> |      Methods       | ratio | BoolQ | PIQA  | HellaSwag | WinoGrande | ARC-e | ARC-c | OBQA  | Average |
> |--------------------|-------|-------|-------|-----------|------------|-------|-------|-------|---------|
> | Vicuna-7B          |  0%   | 75.69 | 77.91 | 71.04     | 67.80      | 68.98 | 40.7 | 42.20 | 63.47   |
> | SLimLLM w/o tune   | 20%   | 74.92 | 76.12 | 67.98     | 65.82      | 67.09 | 39.33 | 42.60 | 61.98   |
> | SLimLLM w/ tune    | 20%   | 76.15 | 76.39 | 69.32     | 64.72      | 68.56 | 39.25 | 41.80 | 62.31   |
>
> Q9: 13B model
>
> A9: Thank you for your suggestion. We have added experiments on LLaMA-13B.
> |      Methods       | ratio | BoolQ | PIQA  | HellaSwag | WinoGrande | ARC-e | ARC-c | OBQA  | Average |
> |--------------------|-------|-------|-------|-----------|------------|-------|-------|-------|---------|
> | Llama-13B          |  0%   | 68.47 | 78.89 | 76.24     | 70.09      | 74.58 | 44.54 | 42.00 | 64.97   |
> | LoRAP w/o tune     | 20%   | 73.94 | 77.31 | 74.93     | 69.69      | 70.79 | 40.44 | 41.40 | 64.07   |
> | SlimLLM w/o tune   | 20%   | 74.13 | 77.53 | 74.73     | 69.30      | 70.45 | 42.32 | 41.00 | 64.21   |
>
> Q10: Latency
>
> A10: The framwork is Vanilla HuggingFace. To test decoding latency, we set the batch size to 8 and measured the latency for generating 256 tokens. The results are as follows:
> |    ratio    | latency(s) |
> |-------------|------------|
> |    0%       | 13.12      |
> |    20%      | 11.48      |
> |    50%      | 9.38       |
>
> Others: Thank you for comments on writing, we will revise it carefully.

---

> > ### Comment · Reviewer_JstE · 2025-04-08
> >
> > Thank you for your detailed rebuttal and the additional experiments. I believe my initial score may have been somewhat harsh, as I was unsure about the competitiveness of your work. However, your thorough responses and the comments from other reviewers have alleviated my concerns.
> >
> > I would like to increase my score from 1 to 3 for the following reasons:
> > * During the re-read, similarity-based MHA head pruning and regression-based performance recovery feel interesting again. Also, the authors describe the greedy search cost for head pruning, which does not seem to take long (A2).
> > * The comparison with FLAP appears to be impressive (A1), and the additional matrices for performance recovery (A4) appear to be integrable, ensuring compatibility with existing serving frameworks.
> > * Furthermore, several ablations strengthen the experimental validation.
> >
> > However, I am still concerned about the settings for r_0 and alpha (A5), i.e., whether these values are applicable to other models, the effectiveness over LoRAP for Llama-2-7B and Llama-1-13B, and a concern that the models examined seem a bit outdated. Releasing the code would be beneficial for verifying reproducibility and facilitating future work.

---

### Official Review · Reviewer_2F2C · 2025-03-10

**Overall Recommendation:** 4

**Summary:**

This paper proposes SlimLLM, a structured pruning method for LLMs that evaluates redundancy via Pearson similarity-driven head pruning and PCA-guided FFN channel pruning. A lightweight linear recalibration reduces post-pruning accuracy loss, while dynamic layer sparsity optimizes resource allocation. Experiments show strong performance on LLaMA models, surpassing prior methods.

**Claims And Evidence:**

The claims made in the paper are well-supported.

**Essential References Not Discussed:**

N/A

**Experimental Designs Or Analyses:**

The experimental designs and analyses appear to be solid. The proposed method is evaluated on various benchmarks and extensive ablation studies are provided.

**Methods And Evaluation Criteria:**

The evaluation criteria and datasets are adequate for this task.

**Other Comments Or Suggestions:**

For LLaMA2-7B, the proposed method obtains slightly poorer performance than prior methods on pruning ratio of 50% with finetuning. More analysis and discussions are encouraged.

**Other Strengths And Weaknesses:**

**Strengths**

1. Eigenvector-guided PCA preserves critical feature directions in FFN layers, surpassing magnitude-only criteria.
2. The proposed method demonstrates stronger performance than prior methods like LoRAP and LLM-Pruner.
3. The linear regression strategy shows great impact for performance recovery, as demonstrated in Table 5.

**Weaknesses**

1.This paper should be carefully proofread. For example, what is $o_{-hi}$ in Figure 1? $S_{-p}$ is not easy to understand in Algorithm 1 and should be replaced by better equation format.

2.What is exactly the latency reported in Table 3? Prefill latency or Decoding latency? More details should be included.

3.The proposed method is evaluated for LLMs. Is it possible that this method could also prune vision-language models?

**Questions For Authors:**

N/A

**Relation To Broader Scientific Literature:**

SlimLLM provides a novel and accurate method for pruning LLMs.

**Theoretical Claims:**

N/A

---

> ### Author Rebuttal · Authors · 2025-04-01
>
> Q1: This paper should be carefully proofread. For example, what is $O_{-h_{i}}$ in Figure 1? $S_{-p}$ is not easy to understand in Algorithm 1 and should be replaced by better equation format.
>
> A1: Thank you for your suggestion. $O_{-h_{i}}$ denotes the output excluding the $i-th$ head. $S_{-p}$ represents the set of unpruned attention head. $S_{-p}$={$head_{i}$, $i \in$ unpruned head index}. The detailed descriptions will be added in the paper.
>
> Q2: What is exactly the latency reported in Table 3? Prefill latency or Decoding latency? More details should be included.
>
> A2: Thank you for your suggestion. We show the prefill latency in Table 3 and we will add details in paper. Additionally, we conducted experiments to measure the decoding latency. The table below shows the latency results for generating 256 tokens with a batch size of 8.
> |    ratio    | #Param   | latency(s) |
> |-------------|----------|------------|
> |    0%       | 6.7B     | 13.12      |
> |    20%      | 5.4B     | 11.48      |
> |    50%      | 3.4B     | 9.38       |
>
> Q3: The proposed method is evaluated for LLMs. Is it possible that this method could also prune vision-language models?
>
> A3: Yes, it is. For VLM (Visual-Language Model) models, the pruning strategy needs to be adapted, such as the proportion of visual tokens to text tokens. We are currently conducting preliminary experiments on Qwen-VL-7B, and the experimental results are shown in the table. We will present more details in our future work.
> |    Datasets         |  prune ratio |  MME    | MMBench_dev_en |
> |---------------------|--------------|---------|----------------|
> |   Qwen-VL-7B        | 0%           |2292.88  |  77.9          |
> | Qwen-VL-7B w/o tune | 20%          |2047.691 |  74.9          |

---

> > ### Comment · Reviewer_2F2C · 2025-04-04
> >
> > The authors totally address my concern, I will raise my score to 4.

---

### Official Review · Reviewer_5cF2 · 2025-03-11

**Overall Recommendation:** 4

**Summary:**

This paper proposes SlimLLM for pruning large language models (LLMs). The method uniquely combines Pearson correlation analysis for attention head redundancy detection with PCA-based directional importance for FFN channel pruning. A lightweight linear calibration technique minimizes post-pruning performance degradation, while layer-specific sparsity allocation leverages input-output alignment. Evaluations on LLaMA-7B and LLaMA2-7B demonstrate significant efficiency gains while maintaining competitive accuracy on commonsense reasoning tasks.

**Claims And Evidence:**

The authors claim SlimLLM achieves state-of-the-art structured pruning for LLMs. This is substantiated by Table 1, where SlimLLM outperforms LoRAP by 2.85% average accuracy at 50% pruning without fine-tuning, and Table 3, which highlights a 3.4× speedup on LLaMA-7B.

**Essential References Not Discussed:**

NA.

**Experimental Designs Or Analyses:**

This paper provides extensive experimental results and ablation studies.

**Methods And Evaluation Criteria:**

The framework is evaluated on various tasks (BoolQ, PIQA etc.) using LLaMA-family models. Latency measurements on NVIDIA V100 GPUs confirm practical applicability.

**Other Comments Or Suggestions:**

See weakness part.

**Other Strengths And Weaknesses:**

**Strengths**
-  PCA-based feature importance for FFN channels addresses a critical limitation of magnitude-only pruning, preserving directional information.
- The proposed method achieves great improvement for running speed, making it viable for real-time applications.
- The linear regression strategy is an interesting idea and recovers performance with negligible computational overhead.
**Weaknesses**
1. Computational complexity of iterative head pruning (Algorithm 1) is unaddressed, raising concerns for larger models.
2. Linear regression is applied only to output matrices. Is it possible for extending it to intermediate layers for further reducing accuracy loss.

**Questions For Authors:**

How does the computational cost of greedy search scale with model size?

**Relation To Broader Scientific Literature:**

LLM pruning is a popular topic and this paper proposes a novel method that achieves strong performance.

**Theoretical Claims:**

None explicitly stated.

---

> ### Author Rebuttal · Authors · 2025-04-01
>
> Q1: Computational complexity of iterative head pruning (Algorithm 1) is unaddressed, raising concerns for larger models.
>
> A1: The computational complexity of Algorithm 1 is acceptable. First, the number of heads is generally small, which limits the number of iterations in the algorithm. Second, the outputs of each head can be precomputed and each iteration only involves the calculation of similarity, which reduces the number of redundant calculations during the iterations. For each layer, the time spent on greedy search is approximately 2 seconds when pruning Llama-7B.
>
> Q2: Linear regression is applied only to output matrices. Is it possible for extending it to intermediate layers for further reducing accuracy loss.
>
> A2: We compensate for the pruning error by performing linear fitting between the outputs before and after pruning. This requires that the outputs of the two stages maintain the same dimensionality. Currently, the output matrices of MHA and FFN meet this requirement. However, for other linear layers, since the output channels have been pruned, this method cannot be directly applied at present.

---

### Official Review · Reviewer_YsF4 · 2025-03-11

**Overall Recommendation:** 3

**Summary:**

This paper proposes a structured pruning method for large language models (LLMs) that compresses both the feed-forward (FFN) and attention layers to accelerate inference. The pruning algorithm incorporates two key techniques: (1) removing redundant attention heads based on Pearson similarity and (2) pruning FFN layers using principal component analysis (PCA) on feature representations. After pruning, the method employs linear regression to efficiently recover the pruned weights by minimizing reconstruction error. The paper evaluates the approach on the LLaMA-7B model, assessing performance across benchmarks including WikiText, PTB, and BoolQ, among others. Experimental results demonstrate that SlimLLM outperforms baselines such as LLM-Pruner, LoRAPrune, and LoRAP.

## update after rebuttal

Thanks for the response. I will keep my initial positive score.

**Claims And Evidence:**

Yes

**Essential References Not Discussed:**

N/A

**Experimental Designs Or Analyses:**

Yes.

**Methods And Evaluation Criteria:**

Yes

**Other Comments Or Suggestions:**

N/A

**Other Strengths And Weaknesses:**

## Strengths:

1. SlimLLM is a simple yet highly practical structured pruning method. It requires no additional training and can be easily applied to various large language models (LLMs), making it a scalable solution for real-world deployment.
2. The proposed fast recovery method achieves performance levels comparable to low-cost fine-tuning. For example, at a 50% pruning rate, the LoRA-based LLM-Pruner attains a perplexity (PPL) of 38.12 on WikiText, while SlimLLM achieves a slightly lower PPL of 37.89, demonstrating its effectiveness in maintaining model accuracy despite significant compression.
3. Unlike uniform pruning approaches, SlimLLM incorporates a non-uniform pruning ratio, dynamically adjusting layerwise pruning based on cosine similarity between input and output representations.

## Weaknesses:

1. Although the proposed sub-methods have been proven effective, the relation between FFN pruning and attention layers pruning is no so clear. Are they independent methods?
2. The paper mainly focuses on LLaMA models. It is encouraged to conduct more experiments on other models.
3. The performance gain appears to be not so significant at a 20% pruning ratio, as shown in Table 5. This suggests that SlimLLM might be more beneficial for higher compression levels but less impactful for moderate pruning. A deeper investigation into why performance varies across different pruning ratios would be valuable.
4. The layer-wise pruning ratio relies on empirical hyperparameter α, lacking theoretical justification and more ablation studies.

**Questions For Authors:**

Please see the weaknesses.

**Relation To Broader Scientific Literature:**

N/A

**Theoretical Claims:**

No theoretical results in this paper.

---

> ### Author Rebuttal · Authors · 2025-04-01
>
> Q1: Although the proposed sub-methods have been proven effective, the relation between FFN pruning and attention layers pruning is no so clear. Are they independent methods?
>
> A1: In this method, both the head pruning and channel pruning strategies are designed to increase the linear correlation between the outputs before and after pruning, which benefits our linear regression strategy. Meanwhile, considering the significant difference in the number of heads and channels, we employ the PCA method in the FFN to make the pruned output as close as possible to the principal direction. In the MHA, we directly maximize the Pearson similarity to bring the similarity closer.
>
> Q2: The paper mainly focuses on LLaMA models. It is encouraged to conduct more experiments on other models.
>
> A2: Thank you for your suggestion. We have added the experimental results on Vicuna-7B.
>
> |      Methods       | ratio | BoolQ | PIQA  | HellaSwag | WinoGrande | ARC-e | ARC-c | OBQA  | Average |
> |--------------------|-------|-------|-------|-----------|------------|-------|-------|-------|---------|
> | Vicuna-7B          |  0%   | 75.69 | 77.91 | 71.04     | 67.80      | 68.98 | 40.7 | 42.20 | 63.47   |
> | LLMPruner w/o tune | 20%   | 62.87 | 75.41 | 64.00     | 58.41      | 60.98 | 37.12 | 39.00 | 56.83   |
> | LLMPruner w/ tune  | 20%   | 60.40 | 75.63 | 65.45     | 63.22      | 63.05 | 37.71 | 39.00 | 57.78   |
> | LoRAP w/o tune     | 20%   | 76.42 | 76.38 | 68.31     | 64.96      | 65.82 | 37.29 | 38.60 | 61.11   |
> | LoRAP w/ tune      | 20%   | 75.81 | 76.77 | 68.39     | 65.04      | 70.08 | 39.33 | 39.20 | 62.09   |
> | SLimLLM w/o tune   | 20%   | 74.92 | 76.12 | 67.98     | 65.82      | 67.09 | 39.33 | 42.60 | 61.98   |
> | SLimLLM w/ tune    | 20%   | 76.15 | 76.39 | 69.32     | 64.72      | 68.56 | 39.25 | 41.80 | 62.31   |
>
>
> Q3: The performance gain appears to be not so significant at a 20% pruning ratio, as shown in Table 5. This suggests that SlimLLM might be more beneficial for higher compression levels but less impactful for moderate pruning. A deeper investigation into why performance varies across different pruning ratios would be valuable.
>
> A3: We posit that the impact on model performance is comparatively minimal at a pruning ratio of 20%, whereas the pruning error becomes significantly more pronounced at a pruning ratio of 50%. Our proposed strategy is capable of effectively mitigating this error, thereby yielding more substantial performance improvements.
>
> Q4: The layer-wise pruning ratio relies on empirical hyperparameter α, lacking theoretical justification and more ablation studies.
>
> A4: Thank you for your suggestion. We have added ablation experiments on Llama-7B, and the results are as follows. When the value of alpha increases to 7, the model achieves optimal performance. Further increment of alpha leads to a decline in model performance.
> |    alpha    | ratio | wikitext2$\downarrow$ |  PTB$\downarrow$  |   Avg.$\uparrow$ |
> |-------------|-------|-----------|-------|-----------------|
> |     1       |  50%  |    54.17  | 96.57 | 44.42           |
> |     4       |  50%  |    43.36  | 83.91 | 44.95           |
> |     7       |  50%  |    37.89  | 67.68 | 50.10           |
> |     10      |  50%  |    44.82  | 75.80 | 46.86           |

---

### Official Review · Reviewer_aFXN · 2025-03-18

**Overall Recommendation:** 4

**Summary:**

The paper presents SlimLLM, a structured pruning approach for large language models (LLMs) that tackles channel and attention head pruning through a unified importance evaluation framework. The paper introduces several novel techniques, including using Pearson similarity to identify redundant attention heads with a greedy search strategy, evaluating FFN channel importance via PCA to prioritize feature directions, and a layerwise pruning strategy that adjusts sparsity based on input-output cosine similarity. Evaluated on LLaMA-7B/2-7B, SlimLLM achieves better accuracy and lower perplexity than baselines like LoRAP.

**Claims And Evidence:**

The claims made in the paper are well-supported by extensive comparisons and ablation studies.

**Essential References Not Discussed:**

NA.

**Experimental Designs Or Analyses:**

The experimental setup is reasonable, with all comparative experiments conducted under the same conditions. Additionally, extensive ablation studies are provided to demonstrate the effectiveness of each proposed module.

**Methods And Evaluation Criteria:**

The proposed method is evaluated on various benchmarks for LLaMA-7B and LLaMA2-7B. Evaluation focuses on zero-shot accuracy across commonsense reasoning datasets (e.g., BoolQ, PIQA), perplexity (WikiText2, PTB), and inference latency.

**Other Comments Or Suggestions:**

NA

**Other Strengths And Weaknesses:**

Strengths

1. SlimLLM introduces novel criteria for structured pruning, such as Pearson similarity-based head pruning and PCA-driven feature space importance for FFN channels. These methods move beyond element-wise aggregation, capturing interdependencies within sub-modules and providing a more comprehensive assessment of redundancy. The greedy search for head combinations further enhances this approach, addressing limitations of prior works like LoRAP that rely on weight magnitude or activation norms.

2. The proposed linear regression strategy for output matrix fine-tuning is both simple and effective. It achieves significant performance recovery with negligible computational cost, avoiding complex retraining.

3. The paper provides comprehensive comparative experiments and ablation studies to justify the effectiveness of the proposed method.

Weaknesses:

1. The paper focuses on pruning-based acceleration of LLMs but lacks a comparison with quantization-based acceleration methods. Including such a comparison would better demonstrate the proposed approach's advantages of LLM compressing techniques.

2. Table 2 only compares the proposed method with LoRAP. Expanding the evaluation to include more state-of-the-art pruning or compression baselines would strengthen the demonstration of the method's effectiveness and generality.

**Questions For Authors:**

NA

**Relation To Broader Scientific Literature:**

The paper focuses on LLM pruning, a critical problem in the field that has garnered significant attention from both the academic community and industry.

**Theoretical Claims:**

The paper does not provide too much theoretical analysis.

---

> ### Author Rebuttal · Authors · 2025-04-01
>
> Q1: The paper focuses on pruning-based acceleration of LLMs but lacks a comparison with quantization-based acceleration methods. Including such a comparison would better demonstrate the proposed approach's advantages of LLM compressing techniques.
>
> A1:  Thank you for your suggestion. Quantization is also a highly effective method for LLM compression. Since the directions of pruning and quantization as compression strategies are different and they can yield cumulative benefits, we have not compared our method with quantization at present.
>
> Q2: Table 2 only compares the proposed method with LoRAP. Expanding the evaluation to include more state-of-the-art pruning or compression baselines would strengthen the demonstration of the method's effectiveness and generality.
>
> A2: Thank you for your suggestion. In Table 2, we compared our method with the LoRAP method, which has excellent compression performance. However, due to memory limitations, we were unable to conduct experiments on LLMPruner with Llama2-7B. Below are the relevant results reproduced from other works, which show that our pruning strategy can better preserve model performance.
>
> |    Methods    | ratio | BoolQ | PIQA  | HellaSwag | WinoGrande | ARC-e | ARC-c | OBQA  | Average |
> |---------------|-------|-------|-------|-----------|------------|-------|-------|-------|---------|
> | LLaMA2-7B     |  0%   | 71.04 | 78.40 | 72.96     | 67.17      | 69.32 | 40.53 | 40.08 | 62.89   |
> | LoRAP         | 20%   | 69.24 | 76.39 | 69.15     | 65.11      | 61.99 | 35.58 | 38.60 | 59.44   |
> | LLMPruner     | 20%   | 67.95 | 77.58 | 71.43     | 64.01      | 63.51 | 38.05 | 39.80 | 60.33   |
> | SLimLLM(ours) | 20%   | 69.79 | 76.28 | 68.88     | 63.54      | 65.74 | 39.08 | 39.80 | 60.44   |

---

### Decision · Program_Chairs · 2025-05-01

**Decision:**

Accept (poster)

**Comment:**

This paper puts forward SlimLLM, a structured pruning approach for large language models (LLMs). It assesses redundancy through Pearson similarity-based head pruning and PCA-directed feed-forward network (FFN) channel pruning. A lightweight linear recalibration is used to decrease the accuracy loss after pruning, and dynamic layer sparsity helps optimize resource distribution. Experiments carried out on LLaMA models demonstrate that SlimLLM performs well and outperforms previous pruning methods.

All reviewers recommend acceptance of this work. All the questions are solved during the discussion phase.

Thus, the AC recommends acceptance of this work as poster.